

# Bootstrapping mixed correlators
# in three-dimensional cubic theories

Stefanos R. Kousvos[1] and Andreas Stergiou[2,3]

**1** ITCP and Department of Physics, University of Crete, 700 13 Heraklion, Greece
**2** Theoretical Physics Department, CERN, 1211 Geneva 23, Switzerland
**3** Theoretical Division, MS B285, Los Alamos National Laboratory,
Los Alamos, NM 87545, USA

## Abstract

Three-dimensional theories with cubic symmetry are studied using the machinery of the numerical conformal bootstrap. Crossing symmetry and unitarity are imposed on a set of mixed correlators, and various aspects of the parameter space are probed for consistency. An isolated allowed region in parameter space is found under certain assumptions involving pushing operator dimensions above marginality, indicating the existence of a conformal field theory in this region. The obtained results have possible applications for ferromagnetic phase transitions as well as structural phase transitions in crystals. They are in tension with previous $\varepsilon$ expansion results, as noticed already in earlier work.

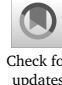
## 1 Introduction

The cubic deformation of the Heisenberg model in three spacetime dimensions is of paramount importance for the critical behavior of systems as simple as magnets like Fe or Ni. In these, as

well as other systems, the cubic deformation is allowed in the context of the Landau theory of phase transitions, and thus its effects need to be taken into account in order to find the fixed point to which the flow is driven at low energies. In the past, this has been addressed mainly with perturbative methods like the $\varepsilon$ expansion [1], while Monte Carlo simulations have been very limited [2]. The objective in those studies was to find out which fixed point is the stable one under particular deformations. In this work we study theories with cubic symmetry using the numerical conformal bootstrap [3]. The cubic group, $C_3$, is a subgroup of the orthogonal group $O(3)$; it can be written as a semi-direct product, $C_3 = \mathbb{Z}_2{}^3 \rtimes S_3$, or a direct product, $C_3 = S_4 \times \mathbb{Z}_2$, where $S_n$ is the symmetric group. In crystallographic notation it is the group $O_h$. Our analysis is based just on the presence of cubic symmetry and unitarity, and does not assume a Landau–Ginzburg description of the fixed point, as is the case when perturbative methods are used.

One of the most widely-used strategies in the numerical conformal bootstrap is to make an assumption about the scaling dimension of an operator (the external operator) and obtain a bound on the scaling dimension of another operator (the exchanged operator) that appears in the operator product expansion (OPE) of the first operator with itself. As in other examples, in the case of theories with cubic symmetry it is natural to first consider the order-parameter operator $\phi_i, i = 1, 2, 3$. This operator has lowest possible dimension $1/2$ consistently with unitarity, and it furnishes a three-dimensional irreducible representation (irrep) of $C_3$. Its OPE with itself takes the schematic form

$$\phi_i \times \phi_j \sim \delta_{ij} S + X_{(ij)} + Y_{(ij)} + A_{[ij]}, \tag{1}$$

where $S$ is the one-dimensional singlet irrep, $X$ a two-dimensional symmetric irrep, $Y$ a three-dimensional symmetric irrep, and $A$ a three-dimensional antisymmetric irrep. When we view $C_3$ as a subgroup of $O(3)$, the irreps $X$ and $Y$ stem from the traceless-symmetric irrep of $O(3)$, which is reducible under the action of $C_3$. There is a $\mathbb{Z}_2$ symmetry under which $\phi_i$ is charged, so the operators in the right-hand side of 1 are all $\mathbb{Z}_2$-even.

In a recent paper by one of the authors [4], a plot on the dimension of the first $X$ operator in the $\phi_i \times \phi_j$ OPE was obtained, which showed a change in slope of the boundary curve; see Fig. 1. Such a feature, commonly referred to as a "kink", has been seen in other examples to appear due to the presence of a known conformal field theory (CFT) at that location in parameter space, e.g. the Ising model [5, 6] and the $O(N)$ models [7]. In other words, the parameters obtained when saturating the kink have been found to be operator dimensions of a CFT. The aim of this paper is to examine the possibility that the kink in Fig. 1 might also correspond to the location of an actual cubic symmetric CFT, and a solution to the crossing equation that is not an artifact of the numerics. Note that a bound on the dimension of the first singlet scalar $S$ in the $\phi \times \phi$ OPE was also obtained in [4], but it was identical to the one obtained in the $O(3)$ case. This limits the utility of that bound, for saturating it puts us on the $O(3)$ solution, with the cubic one somewhere in the allowed region. However, the coincidence of the bounds still carries some useful information, namely that the first singlet scalar in cubic theories has dimension lower than that in theories with $O(3)$ symmetry.

In Fig. 1 we point out the position of the decoupled Ising model, a known CFT with cubic symmetry that lies in the allowed region of the bound.[1] We would like to emphasize here that this theory, in which $\Delta_X = \Delta_\epsilon \approx 1.4126$, where $\epsilon$ is the first $\mathbb{Z}_2$-even scalar operator in the Ising model, does not saturate the bound. Although it lies very close to it, its distance from the bound is numerically significant; see [4, Fig. 6]. The bound presented in Fig. 1 has essentially converged to the optimal one, as was verified in [4] by increasing the numerical

---

[1]The decoupled Ising model arises simply by taking $N$ copies of the Ising model. In our case $N = 3$. Each Ising model has a $\mathbb{Z}_2$ symmetry and permuting the Ising models results in the group $C_N = \mathbb{Z}_2{}^N \rtimes S_N$ for the decoupled Ising theory.

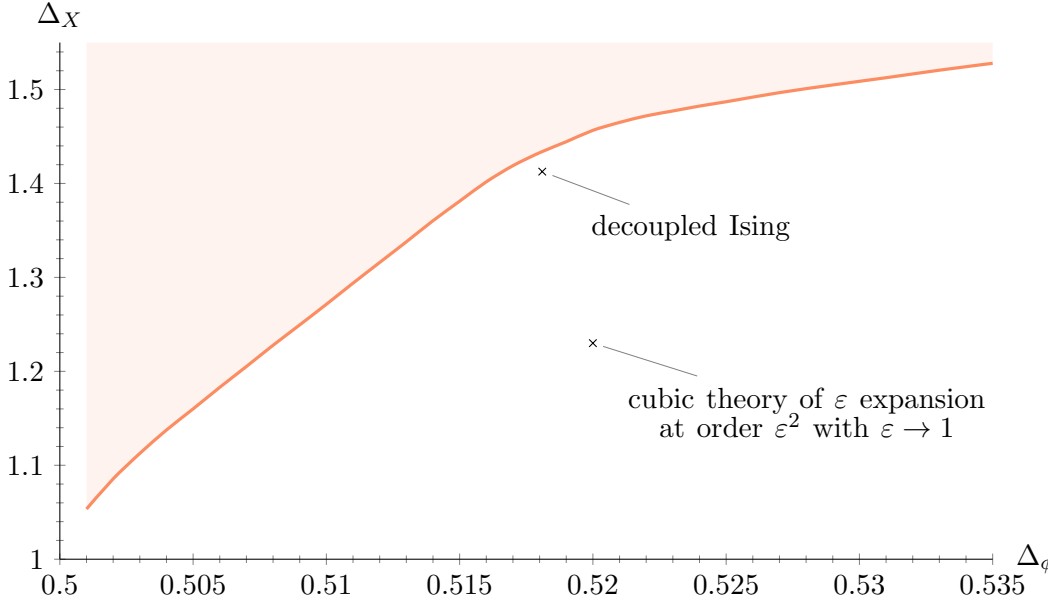

Figure 1: Upper bound on the dimension of the first $X$ operator in the $\phi_i \times \phi_j$ OPE. The red area is excluded.

complexity of the algorithms and observing that the bound did not get stronger. Furthermore, an analysis of the spectrum along the bound yielded results inconsistent with the spectrum of the decoupled Ising model. We also include the location of the cubic theory of the $\varepsilon$ expansion at order $\varepsilon^2$ using results of [8]. Assuming that higher orders and resummations do not change this location significantly, our assumption below that $\Delta_X$ lies on the bound of Fig. 1 excludes from our subsequent plots the possibility that our cubic theory is that of the $\varepsilon$ expansion.

Besides cubic magnets, CFTs with cubic symmetry have potential relevance for structural phase transitions [9–11]. These are continuous phase transitions in which the crystallographic structure of a crystal changes at a specific temperature, with the high-temperature, undistorted phase having a symmetry that is broken in the low-temperature, distorted phase. In the cubic-to-tetragonal phase transition of $SrTiO_3$ (strontium titanate) [12–16], whose perovskite structure is seen in Fig. 2, the situation is described in Fig. 3; the cubic symmetry of the undis-

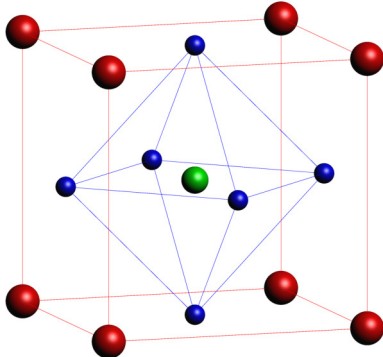

Figure 2: The perovskite structure of $SrTiO_3$. Sr is red, Ti is green, and $O_3$ is blue.

torted phase is reduced due to the transition to the tetragonal crystallographic system below the critical temperature. Since the cubic deformation is allowed in the undistorted phase, CFTs with cubic symmetry and three-dimensional order parameters may be relevant for the critical behavior of cubic systems that undergo structural phase transitions.

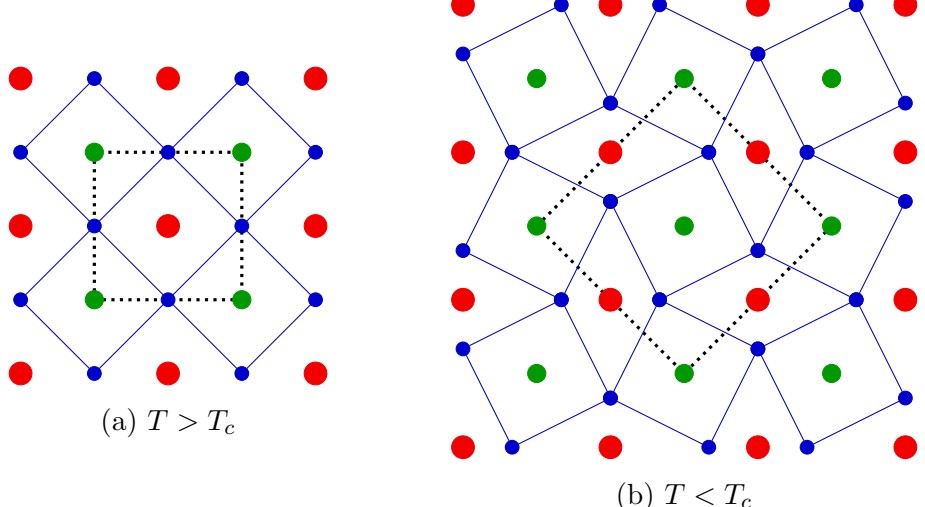

Figure 3: The crystallographic structure of $SrTiO_3$ in a top-down view above (a) and below (b) the critical transition temperature $T_c \approx 100$ K. The unit cell is highlighted by the dotted line. The unit cell of the distorted phase is enlarged by $\sqrt{2} \times \sqrt{2} \times 2$ relative to the undistorted phase, since two oxygen octahedra on top of each other rotate in opposite directions. The crystal system is cubic in the undistorted and tetragonal in the distorted phase.

With these motivations in mind we undertake in this work the study of a system of correlation functions involving $\phi$ and $X$ using the numerical bootstrap. More specifically, we analyze crossing and unitarity constraints on the correlators $\langle \phi\phi\phi\phi \rangle$, $\langle \phi\phi XX \rangle$ and $\langle XXXX \rangle$. We assume throughout that the dimension of $X$ saturates the bound of Fig. 1. Our considerations follow the logic described in [17, 18].

When considering mixed correlators one has to analyze more OPEs besides 1. In our case this consists of the OPEs $\phi \times X$ and $X \times X$. The group theory required to understand these OPEs as well as the decomposition of the various four-point functions under the cubic group will be presented in detail below. Crossing symmetry leads to a system of thirteen crossing equations which are analyzed using standard algorithms [19, 20].

Our numerical results show that there exists an isolated region in parameter space, consistent with crossing and unitarity, obtained by making assumptions of irrelevance (in the RG sense) of the second operators in the singlet and two-dimensional irreps, called $S'$ and $X'$, respectively. Note that these operators also appear in the OPE $\phi_i \times \phi_j$ analyzed in [4], but crossing symmetry imposed only on the correlator $\langle \phi\phi\phi\phi \rangle$ is not enough to give us the isolated region found with the mixed correlators. As we will see, the essential extra constraint used in the mixed-correlator bootstrap is the equality of certain OPE coefficients. The importance of using equality of OPE coefficients has already been seen in the case of the $O(N)$ models [18].

Our results for the critical exponents $\beta = \Delta_\phi/(3-\Delta_S)$ and $\nu = 1/(3-\Delta_S)$ in the obtained isolated allowed region, where $\Delta_\phi = 0.518 \pm 0.001$ and $\Delta_S = 1.317 \pm 0.012$, are

$$\beta = 0.308 \pm 0.002\,, \qquad \nu = 0.594 \pm 0.004\,, \tag{2}$$

Based on 2 we suggest that there exists a previously-unknown CFT that is relevant for structural phase transitions, as described above, where [12] and [14] give the measurements

$$\beta = 0.33 \pm 0.02\,, \qquad \nu = 0.63 \pm 0.07\,, \tag{3}$$

respectively. Our suggestion is also based on the presence of cubic symmetry and a three-dimensional order parameter. The critical exponent $\beta$ in 3 has also been reported in the ferromagnetic phase transition of EuS (europium sulfide) [21]. Perhaps other physical systems belong to the same universality class. Experiments that can shrink the critical exponents' error margins would be crucial in testing our suggestion.

This paper is organized as follows. In the next section we describe in detail results for the OPEs and four-point functions that are necessary for our analysis, and list the final crossing equation we use in our numerical explorations. In section 3 we present our results, and we conclude in section 4.

## 2 OPEs, four-point functions, and crossing equations

In this section we will analyze in detail the OPEs and the four-point functions we use in our bootstrap analysis. The consequences of cubic symmetry for OPEs and four-point functions of operators transforming in irreps of $C_3$ have not been explored very much in the literature, so we will attempt to provide a self-contained treatment.

The group $C_3$ has ten irreps. Viewed as $S_4 \times \mathbb{Z}_2$, these are the five irreps of $S_4$ for each parity, namely the $\mathbf{1}$ (singlet), the $\bar{\mathbf{1}}$ (antisinglet), the $\mathbf{2}$ (diagonal), the $\mathbf{3}$ (off-diagonal), and the $\bar{\mathbf{3}}$ (antisymmetric). These irreps are nicely described by the Young tableaux (see e.g. [22])

$$\mathbf{1}: \square\square\square\square \,, \quad \bar{\mathbf{1}}: \begin{array}{c}\square\\\square\\\square\\\square\end{array} \,, \quad \mathbf{2}: \begin{array}{c}\square\square\\\square\square\end{array} \,, \quad \mathbf{3}: \begin{array}{c}\square\square\square\\\square\end{array} \,, \quad \bar{\mathbf{3}}: \begin{array}{c}\square\square\\\square\\\square\end{array} \,. \tag{4}$$

The names diagonal and off-diagonal for the $\mathbf{2}$ and the $\mathbf{3}$, respectively, stem from the location of the entries that make them up in the traceless symmetric irrep of $O(3)$ from which they descend. The traceless symmetric irrep of $O(3)$ is not irreducible under the action of $C_3$, but splits into diagonal elements making up the $\mathbf{2}$ and off-diagonal elements making up the $\mathbf{3}$. The antisinglet irrep $\bar{\mathbf{1}}$ of $C_3$ is an independent antisymmetric one-dimensional irrep.

### 2.1 OPEs

The order-parameter operator $\phi_i, i = 1, 2, 3$ belongs to the off-diagonal irrep and is $\mathbb{Z}_2$-odd. Its OPE with itself takes the form

$$\phi_i \times \phi_j \sim \delta_{ij} S^+ + X_{ij}^+ + Y_{ij}^+ + A_{ij}^- \,, \tag{5}$$

where $S$ is in the singlet, $X_{ij}$ in the diagonal, $Y_{ij}$ in the off-diagonal, and $A_{ij}$ in the antisymmetric irrep of $C_3$. $X_{ij}$ and $Y_{ij}$ are symmetric. The signs in the superscripts indicate the spin with which these operators appear in the OPE: even (+) or odd (−).

Let us note here that the off-diagonal irrep can be furnished by an operator with one or with two indices. More specifically, one can write

$$Y_{ij} = \gamma_{ijk} Z_k \,, \tag{6}$$

where $\gamma_{ijk}$ is a symmetric tensor with

$$\gamma_{123} = \tfrac{1}{\sqrt{2}} \tag{7}$$

and all other independent components equal to zero. It exists only for the group $C_3$[2] and not for the hypercubic groups $C_{N>3}$. Note that both $Y$ and $Z$ in 6 have the same $\mathbb{Z}_2$ parity. One

---

[2]Strictly speaking, it exists for $S_4$, not $S_4 \times \mathbb{Z}_2 = C_3$.

can verify that

$$\gamma_{ijm}\gamma_{klm} = -\delta_{ijkl} + \tfrac{1}{2}(\delta_{ik}\delta_{jl} + \delta_{il}\delta_{jk}), \tag{8}$$

where $\delta_{ijkl}$ is one if $i = j = k = l$ and zero otherwise.

They are perhaps unfamiliar, so it may be useful to give here the global symmetry structure of the two-point functions of operators in the diagonal and off-diagonal irreps of $C_3$. For completeness, we include the global symmetry structure of the two-point function of antisymmetric irreps:

$$\langle X_{ij}X_{kl}\rangle \sim \delta_{ijkl} - \tfrac{1}{3}\delta_{ij}\delta_{kl}, \tag{9}$$

$$\langle Y_{ij}Y_{kl}\rangle \sim -\delta_{ijkl} + \tfrac{1}{2}(\delta_{ik}\delta_{jl} + \delta_{il}\delta_{jk}), \tag{10}$$

$$\langle A_{ij}A_{kl}\rangle \sim -\tfrac{1}{2}(\delta_{ik}\delta_{jl} - \delta_{il}\delta_{jk}). \tag{11}$$

As we see $\langle Y_{ij}Y_{kl}\rangle = \gamma_{ijm}\gamma_{kln}\langle Z_m Z_n\rangle \sim \gamma_{ijm}\gamma_{kln}\delta_{mn}$, which then correctly reproduces 10 due to 8.

Another OPE we need for our analysis is that of $\phi_i$ with $X_{jk}$, which takes the form

$$\phi_i \times X_{jk} \sim (\delta_{ijkl} - \tfrac{1}{3}\delta_{il}\delta_{jk})Y_l^{\prime\pm} + \delta_{jkl[m}\gamma_{n]li}A_{mn}^{\prime\pm}. \tag{12}$$

The operators in the right-hand side of 12 are $\mathbb{Z}_2$-odd and $Y'$ transforms in the off-diagonal irrep.[3] Finally, we need the OPE of $X_{ij}$ with itself. This can be written in the form

$$X_{ij} \times X_{kl} \sim (\delta_{ijkl} - \tfrac{1}{3}\delta_{ij}\delta_{kl})S^+ + \zeta_{ijkl}\bar{S}^-$$
$$+ (\delta_{ijklmn} - \tfrac{1}{3}(\delta_{ij}\delta_{klmn} + \delta_{kl}\delta_{ijmn}) + \tfrac{1}{9}\delta_{ij}\delta_{kl}\delta_{mn})X_{mn}^+, \tag{13}$$

where $\delta_{ijklmn}$ is one if $i = j = k = l = m = n$ and zero otherwise, and

$$\zeta_{ijkl} = \delta_{i1}\delta_{j1}(\delta_{k2}\delta_{l2} - \delta_{k3}\delta_{l3}) - \delta_{i2}\delta_{j2}(\delta_{k1}\delta_{l1} - \delta_{k3}\delta_{l3}) + \delta_{i3}\delta_{j3}(\delta_{k1}\delta_{l1} - \delta_{k2}\delta_{l2}), \tag{14}$$

which is traceless in $i, j$ and $k, l$ and antisymmetric under $ij \leftrightarrow kl$. The operators in the right-hand side of 13 are $\mathbb{Z}_2$-even. The tensor in the last term in 13 comes from $(\delta_{ijmn} - \tfrac{1}{3}\delta_{ij}\delta_{mn})(\delta_{klmp} - \tfrac{1}{3}\delta_{kl}\delta_{mp})X_{np}^+$. Note that, although not necessary, we keep the $\tfrac{1}{9}\delta_{ij}\delta_{kl}\delta_{mn}$ contribution despite the fact that $\delta_{mn}X_{mn}^+ = 0$. This way the tensor is traceless in $i, j$ and $k, l$ even before we use the tracelessness of $X_{mn}^+$.

## 2.2 Four-point functions

With the OPEs 5, 12 and 13 in hand we can now proceed to the analysis of the four-point functions relevant for our bootstrap analysis. The strategy we employ is to expand the four-point functions in a basis of linearly independent invariant projectors. Since we know all required OPEs, this is a simple exercise, although sufficient care is required in order to identify relations among particular combinations of tensors so that we end up with linearly-independent crossing equations. We will present our results in the $12 \to 34$ channel.

The four-point function $\langle \phi_i \phi_j \phi_k \phi_l \rangle$ has already been analyzed in detail in [4,23]. We have

---

[3]We hope the notation with the primes indicating these $\mathbb{Z}_2$-odd irreps will not cause confusion with the earlier notation of primes indicating the second operators in particular irreps.

$$x_{12}^{2\Delta_\phi} x_{34}^{2\Delta_\phi} \langle \phi_i(x_1)\phi_j(x_2)\phi_k(x_3)\phi_l(x_4) \rangle = \sum_{S^+} \lambda_{\phi\phi\mathcal{O}_S}^2 P_{1\,ijkl}^{\phi\phi;\phi\phi} g_{\Delta,\ell}^{\phi\phi;\phi\phi}(u,v)$$

$$+ \sum_{X^+} \lambda_{\phi\phi\mathcal{O}_X}^2 P_{2\,ijkl}^{\phi\phi;\phi\phi} g_{\Delta,\ell}^{\phi\phi;\phi\phi}(u,v) + \sum_{Y^+} \lambda_{\phi\phi\mathcal{O}_Y}^2 P_{3\,ijkl}^{\phi\phi;\phi\phi} g_{\Delta,\ell}^{\phi\phi;\phi\phi}(u,v)$$

$$- \sum_{A^-} \lambda_{\phi\phi\mathcal{O}_A}^2 P_{4\,ijkl}^{\phi\phi;\phi\phi} g_{\Delta,\ell}^{\phi\phi;\phi\phi}(u,v), \tag{15}$$

where we use the conventions of [19] for the conformal block, and the projectors are given by

$$P_{1\,ijkl}^{\phi\phi;\phi\phi} = \delta_{ij}\delta_{kl}, \qquad P_{2\,ijkl}^{\phi\phi;\phi\phi} = \delta_{ijkl} - \tfrac{1}{3}\delta_{ij}\delta_{kl},$$
$$P_{3\,ijkl}^{\phi\phi;\phi\phi} = -\delta_{ijkl} + \tfrac{1}{2}(\delta_{ik}\delta_{jl} + \delta_{il}\delta_{jk}), \qquad P_{4\,ijkl}^{\phi\phi;\phi\phi} = -(\delta_{ik}\delta_{jl} - \delta_{il}\delta_{jk}). \tag{16}$$

Note that, strictly speaking, projectors should satisfy

$$P_{I\,ijmn}P_{J\,nmkl} = P_{I\,ijkl}\,\delta_{IJ}, \qquad \sum_I P_{I\,ijkl} = \delta_{il}\delta_{jk}, \qquad P_{I\,ijkl}\,\delta_{il}\delta_{jk} = d_r^{(I)}, \tag{17}$$

where $d_r^{(I)}$ is the dimension of the representation indexed by $I$. However, the tensors in 16 have been rescaled by positive factors that have been absorbed into the corresponding OPE coefficients in 15, so 17 are not satisfied by 16 without restoring the appropriate normalizations.

For the four-point function $\langle \phi_i \phi_j X_{kl} X_{mn} \rangle$ we find

$$x_{12}^{2\Delta_\phi} x_{34}^{2\Delta_X} \langle \phi_i(x_1)\phi_j(x_2)X_{kl}(x_3)X_{mn}(x_4) \rangle = \sum_{S^+} \lambda_{\phi\phi\mathcal{O}_S}\lambda_{XX\mathcal{O}_S} P_{1\,ijklmn}^{\phi\phi;XX} g_{\Delta,\ell}^{\phi\phi;XX}(u,v)$$

$$+ \sum_{X^+} \lambda_{\phi\phi\mathcal{O}_X}\lambda_{XX\mathcal{O}_X} P_{2\,ijklmn}^{\phi\phi;XX} g_{\Delta,\ell}^{\phi\phi;XX}(u,v), \tag{18}$$

where

$$P_{1\,ijklmn}^{\phi\phi;XX} = \delta_{ij}(\delta_{klmn} - \tfrac{1}{3}\delta_{kl}\delta_{mn}), \tag{19}$$
$$P_{2\,ijklmn}^{\phi\phi;XX} = \delta_{ijklmn} - \tfrac{1}{3}(\delta_{ij}\delta_{klmn} + \delta_{kl}\delta_{ijmn} + \delta_{mn}\delta_{ijkl}) + \tfrac{2}{9}\delta_{ij}\delta_{kl}\delta_{mn}. \tag{20}$$

The four-point function $\langle \phi_i X_{jk}\phi_l X_{mn} \rangle$ takes the form

$$(x_{12}x_{34})^{\Delta_\phi+\Delta_X}\left(\frac{x_{13}}{x_{24}}\right)^{\Delta_\phi-\Delta_X} \langle \phi_i(x_1)X_{jk}(x_2)\phi_l(x_3)X_{mn}(x_4) \rangle =$$

$$\sum_{Y'^\pm} \lambda_{\phi X\mathcal{O}_{Y'}}^2 P_{1\,ijklmn}^{\phi X;\phi X} g_{\Delta,\ell}^{\phi X;\phi X}(u,v) + \sum_{A'^\pm} \lambda_{\phi X\mathcal{O}_{A'}}^2 P_{2\,ijklmn}^{\phi X;\phi X} g_{\Delta,\ell}^{\phi X;\phi X}(u,v), \tag{21}$$

where

$$P_{1\,ijklmn}^{\phi X;\phi X} = \delta_{ijklmn} - \tfrac{1}{3}(\delta_{kl}\delta_{ijmn} + \delta_{mn}\delta_{ijkl}) + \tfrac{1}{9}\delta_{ij}\delta_{kl}\delta_{mn}, \tag{22}$$
$$P_{2\,ijklmn}^{\phi X;\phi X} = -\delta_{ijklmn} + \tfrac{1}{3}(2\delta_{ij}\delta_{klmn} + \delta_{kl}\delta_{ijmn} + \delta_{mn}\delta_{ijkl}) - \tfrac{1}{3}\delta_{ij}\delta_{kl}\delta_{mn}. \tag{23}$$

Note, here, the relation

$$\gamma_{ijk}\gamma_{lmn} = 6\delta_{ijklmn} - 3\,\mathrm{Sym}_{ijk}\,\mathrm{Sym}_{lmn}(3\delta_{il}\delta_{jkmn} - \delta_{il}\delta_{jm}\delta_{kn}), \tag{24}$$

where $\mathrm{Sym}_{i_1\dots i_n}$ symmetrizes in $i_1,\dots,i_n$ and divides by $n!$.

We also note here the result for the four-point function $\langle \phi_i X_{jk} X_{lm} \phi_n \rangle$, which takes the form

$$(x_{12} x_{34})^{\Delta_\phi + \Delta_X} \left( \frac{x_{14}^2}{x_{13} x_{24}} \right)^{\Delta_\phi - \Delta_X} \langle \phi_i(x_1) X_{jk}(x_2) X_{lm}(x_3) \phi_n(x_4) \rangle =$$
$$\sum_{Y'^\pm} \lambda^2_{\phi X \mathcal{O}_{Y'}} (-1)^\ell P^{\phi X; X\phi}_{1\, ijklmn} g^{\phi X; X\phi}_{\Delta, \ell}(u,v) + \sum_{A'^\pm} \lambda^2_{\phi X \mathcal{O}_{A'}} (-1)^\ell P^{\phi X; X\phi}_{2\, ijklmn} g^{\phi X; X\phi}_{\Delta, \ell}(u,v), \quad (25)$$

where

$$P^{\phi X; X\phi}_{1\, ijklmn} = P^{\phi X; \phi X}_{1\, ijknlm}, \qquad P^{\phi X; X\phi}_{2\, ijklmn} = P^{\phi X; \phi X}_{2\, ijknlm}. \quad (26)$$

Finally, for the four-point function $\langle X_{ij} X_{kl} X_{mn} X_{pq} \rangle$ we may write

$$x_{12}^{2\Delta_X} x_{34}^{2\Delta_X} \langle X_{ij}(x_1) X_{kl}(x_2) X_{mn}(x_3) X_{pq}(x_4) \rangle = \sum_{S^+} \lambda^2_{XX \mathcal{O}_S} P^{XX;XX}_{1\, ijklmnpq} g^{XX;XX}_{\Delta, \ell}(u,v)$$
$$- \sum_{\bar{S}^-} \lambda^2_{XX \mathcal{O}_{\bar S}} P^{XX;XX}_{2\, ijklmnpq} g^{XX;XX}_{\Delta, \ell}(u,v) + \sum_{X^+} \lambda^2_{\phi\phi \mathcal{O}_X} P^{XX;XX}_{3\, ijklmn} g^{XX;XX}_{\Delta, \ell}(u,v), \quad (27)$$

where

$$P^{XX;XX}_{1\, ijklmnpq} = (\delta_{ijkl} - \tfrac{1}{3} \delta_{ij} \delta_{kl})(\delta_{mnpq} - \tfrac{1}{3} \delta_{mn} \delta_{pq}), \quad (28)$$

$$P^{XX;XX}_{2\, ijklmnpq} = -\tfrac{1}{3} \zeta_{ijkl} \zeta_{mnpq} = -\delta_{ijmn} \delta_{klpq} + \tfrac{1}{3}(\delta_{ij} \delta_{mn} \delta_{klpq} + \delta_{kl} \delta_{pq} \delta_{ijmn}) - (mn \leftrightarrow pq), \quad (29)$$

$$P^{XX;XX}_{3\, ijklmnpq} = -\delta_{ijkl} \delta_{mnpq} + \delta_{ijmn} \delta_{klpq} + \delta_{ijpq} \delta_{klmn} + \tfrac{1}{3}(\delta_{ij} \delta_{kl} \delta_{mnpq} + \delta_{mn} \delta_{pq} \delta_{ijkl}) \quad (30)$$
$$- \tfrac{1}{3}(\delta_{ij} \delta_{mn} \delta_{klpq} + \delta_{kl} \delta_{pq} \delta_{ijmn}) - \tfrac{1}{3}(\delta_{ij} \delta_{pq} \delta_{klmn} + \delta_{kl} \delta_{mn} \delta_{ijpq}) + \tfrac{1}{9} \delta_{ij} \delta_{kl} \delta_{mn} \delta_{pq}.$$

Note that the tensor $\delta_{ijklmnpq}$, which is one if $i = j = k = l = m = n = p = q$ and zero otherwise, is not independent:

$$\delta_{ijklmnpq} = \tfrac{1}{3}(\delta_{ij} \delta_{klmnpq} + \delta_{kl} \delta_{ijmnpq} + \delta_{mn} \delta_{ijklpq} + \delta_{pq} \delta_{ijklmn})$$
$$+ \tfrac{1}{6}(\delta_{ijkl} \delta_{mnpq} + \delta_{ijmn} \delta_{klpq} + \delta_{ijpq} \delta_{klmn})$$
$$- \tfrac{1}{6}(\delta_{ij} \delta_{kl} \delta_{mnpq} + \delta_{mn} \delta_{pq} \delta_{ijkl}) - \tfrac{1}{6}(\delta_{ij} \delta_{mn} \delta_{klpq} + \delta_{kl} \delta_{pq} \delta_{ijmn})$$
$$- \tfrac{1}{6}(\delta_{ij} \delta_{pq} \delta_{klmn} + \delta_{kl} \delta_{pq} \delta_{ijpq}) + \tfrac{1}{6} \delta_{ij} \delta_{kl} \delta_{mn} \delta_{pq}. \quad (31)$$

We emphasize that equations like 31 are only valid for $N = 3$.

## 2.3 Crossing equations

We can now impose crossing symmetry on the four-point functions involving $\phi$ and $X$ analyzed in the previous subsection. Recall that there the four-point functions were decomposed in the $12 \to 34$ channel, so crossing symmetry requires equating those results with the decomposition of the same four-point functions in the $14 \to 32$ channel. When the dust settles we find thirteen linearly-independent crossing equations. They can be brought to the form

$$\sum_{S^+} (\lambda_{\phi\phi\mathcal{O}_S} \lambda_{XX\mathcal{O}_S}) \vec{T}_{S, \Delta, \ell} \begin{pmatrix} \lambda_{\phi\phi\mathcal{O}_S} \\ \lambda_{XX\mathcal{O}_S} \end{pmatrix} + \sum_{X^+} (\lambda_{\phi\phi\mathcal{O}_X} \lambda_{XX\mathcal{O}_X}) \vec{T}_{X, \Delta, \ell} \begin{pmatrix} \lambda_{\phi\phi\mathcal{O}_X} \\ \lambda_{XX\mathcal{O}_X} \end{pmatrix}$$
$$+ \sum_{Y^+} \lambda^2_{\phi\phi\mathcal{O}_Y} \vec{V}_{Y, \Delta, \ell} + \sum_{A^-} \lambda^2_{\phi\phi\mathcal{O}_A} \vec{V}_{A, \Delta, \ell} + \sum_{Y'^\pm} \lambda^2_{\phi X \mathcal{O}_{Y'}} \vec{V}_{Y', \Delta, \ell} + \sum_{A'^\pm} \lambda^2_{\phi X \mathcal{O}_{A'}} \vec{V}_{A', \Delta, \ell} + \sum_{\bar{S}^-} \lambda^2_{\phi\phi\mathcal{O}_{\bar S}} \vec{V}_{\bar S, \Delta, \ell} = 0, \quad (32)$$

where $\vec{V}_{Y,\Delta,\ell}$, $\vec{V}_{A,\Delta,\ell}$, $\vec{V}_{Y',\Delta,\ell}$, $\vec{V}_{A',\Delta,\ell}$, and $\vec{V}_{\tilde{S},\Delta,\ell}$ are 13-vectors of scalar quantities, while $\vec{T}_{S,\Delta,\ell}$ and $\vec{T}_{X,\Delta,\ell}$ are 13-vectors of $2 \times 2$ matrices. Their components are given by

$$
T^1_{S,\Delta,\ell} = \begin{pmatrix} 0 & 0 \\ 0 & 0 \end{pmatrix}, \quad
T^2_{S,\Delta,\ell} = \begin{pmatrix} F^{\phi\phi;\phi\phi}_{-,\Delta,\ell} & 0 \\ 0 & 0 \end{pmatrix}, \quad
T^3_{S,\Delta,\ell} = \begin{pmatrix} F^{\phi\phi;\phi\phi}_{+,\Delta,\ell} & 0 \\ 0 & 0 \end{pmatrix}, \quad
T^4_{S,\Delta,\ell} = \begin{pmatrix} F^{\phi\phi;\phi\phi}_{-,\Delta,\ell} & 0 \\ 0 & 0 \end{pmatrix},
$$

$$
T^5_{S,\Delta,\ell} = \begin{pmatrix} 0 & 0 \\ 0 & 0 \end{pmatrix}, \quad
T^6_{S,\Delta,\ell} = \begin{pmatrix} 0 & 0 \\ 0 & F^{XX;XX}_{-,\Delta,\ell} \end{pmatrix}, \quad
T^7_{S,\Delta,\ell} = \begin{pmatrix} 0 & 0 \\ 0 & F^{XX;XX}_{+,\Delta,\ell} \end{pmatrix},
$$

$$
T^8_{S,\Delta,\ell} = \begin{pmatrix} 0 & \frac{1}{2}F^{\phi\phi;XX}_{+,\Delta,\ell} \\ \frac{1}{2}F^{\phi\phi;XX}_{+,\Delta,\ell} & 0 \end{pmatrix}, \quad
T^9_{S,\Delta,\ell} = \begin{pmatrix} 0 & \frac{1}{2}F^{\phi\phi;XX}_{+,\Delta,\ell} \\ \frac{1}{2}F^{\phi\phi;XX}_{+,\Delta,\ell} & 0 \end{pmatrix}, \quad
T^{10-13}_{S,\Delta,\ell} = \begin{pmatrix} 0 & 0 \\ 0 & 0 \end{pmatrix},
\tag{33}
$$

$$
T^1_{X,\Delta,\ell} = \begin{pmatrix} 0 & 0 \\ 0 & 0 \end{pmatrix}, \;
T^2_{X,\Delta,\ell} = \begin{pmatrix} -\frac{1}{3}F^{\phi\phi;\phi\phi}_{-,\Delta,\ell} & 0 \\ 0 & 0 \end{pmatrix}, \;
T^3_{X,\Delta,\ell} = \begin{pmatrix} -\frac{1}{3}F^{\phi\phi;\phi\phi}_{+,\Delta,\ell} & 0 \\ 0 & 0 \end{pmatrix}, \;
T^4_{X,\Delta,\ell} = \begin{pmatrix} \frac{2}{3}F^{\phi\phi;\phi\phi}_{-,\Delta,\ell} & 0 \\ 0 & 0 \end{pmatrix},
$$

$$
T^5_{X,\Delta,\ell} = \begin{pmatrix} 0 & 0 \\ 0 & F^{XX;XX}_{-,\Delta,\ell} \end{pmatrix}, \quad
T^6_{X,\Delta,\ell} = \begin{pmatrix} 0 & 0 \\ 0 & 0 \end{pmatrix}, \quad
T^7_{X,\Delta,\ell} = \begin{pmatrix} 0 & 0 \\ 0 & -2F^{XX;XX}_{+,\Delta,\ell} \end{pmatrix},
$$

$$
T^8_{X,\Delta,\ell} = \tfrac{1}{3}T^{10}_{X,\Delta,\ell} = \begin{pmatrix} 0 & \frac{1}{6}F^{\phi\phi;XX}_{-,\Delta,\ell} \\ \frac{1}{6}F^{\phi\phi;XX}_{-,\Delta,\ell} & 0 \end{pmatrix}, \quad
T^9_{X,\Delta,\ell} = \tfrac{1}{3}T^{11}_{X,\Delta,\ell} = \begin{pmatrix} 0 & \frac{1}{6}F^{\phi\phi;XX}_{+,\Delta,\ell} \\ \frac{1}{6}F^{\phi\phi;XX}_{+,\Delta,\ell} & 0 \end{pmatrix},
$$

$$
T^{12,13}_{X,\Delta,\ell} = \begin{pmatrix} 0 & 0 \\ 0 & 0 \end{pmatrix},
\tag{34}
$$

$$
V^1_{Y,\Delta,\ell} = F^{\phi\phi;\phi\phi}_{-,\Delta,\ell}, \quad
V^2_{Y,\Delta,\ell} = F^{\phi\phi;\phi\phi}_{-,\Delta,\ell}, \quad
V^3_{Y,\Delta,\ell} = -F^{\phi\phi;\phi\phi}_{+,\Delta,\ell}, \quad
V^{4-13}_{Y,\Delta,\ell} = 0,
\tag{35}
$$

$$
V^1_{A,\Delta,\ell} = F^{\phi\phi;\phi\phi}_{-,\Delta,\ell}, \quad
V^2_{A,\Delta,\ell} = -F^{\phi\phi;\phi\phi}_{-,\Delta,\ell}, \quad
V^3_{A,\Delta,\ell} = F^{\phi\phi;\phi\phi}_{+,\Delta,\ell}, \quad
V^{4-13}_{A,\Delta,\ell} = 0,
\tag{36}
$$

$$
V^{1-7}_{Y',\Delta,\ell} = 0, \quad
V^8_{Y',\Delta,\ell} = \tfrac{2}{3}V^{10}_{Y',\Delta,\ell} = \tfrac{2}{3}(-1)^\ell F^{\phi X;X\phi}_{-,\Delta,\ell}, \quad
V^9_{Y',\Delta,\ell} = \tfrac{2}{3}V^{11}_{Y',\Delta,\ell} = \tfrac{2}{3}(-1)^{\ell+1} F^{\phi X;X\phi}_{+,\Delta,\ell},
$$

$$
V^{12}_{Y',\Delta,\ell} = F^{\phi X;\phi X}_{-,\Delta,\ell}, \quad
V^{13}_{Y',\Delta,\ell} = 0,
\tag{37}
$$

$$
V^{1-9}_{A',\Delta,\ell} = 0, \quad
V^{10}_{A',\Delta,\ell} = (-1)^{\ell+1} F^{\phi X;X\phi}_{-,\Delta,\ell}, \quad
V^{11}_{A',\Delta,\ell} = (-1)^\ell F^{\phi X;X\phi}_{+,\Delta,\ell},
$$

$$
V^{12}_{A',\Delta,\ell} = 0, \quad
V^{13}_{A',\Delta,\ell} = F^{\phi X;\phi X}_{+,\Delta,\ell},
\tag{38}
$$

and

$$
V^{1-4}_{\tilde{S},\Delta,\ell} = 0, \quad
V^5_{\tilde{S},\Delta,\ell} = -V^6_{\tilde{S},\Delta,\ell} = F^{XX;XX}_{-,\Delta,\ell}, \quad
V^7_{\tilde{S},\Delta,\ell} = F^{XX;XX}_{+,\Delta,\ell}, \quad
V^{8-13}_{\tilde{S},\Delta,\ell} = 0.
\tag{39}
$$

In these equations we use the standard notation

$$
F^{\mathcal{O}_1\mathcal{O}_2;\mathcal{O}_3\mathcal{O}_4}_{\pm,\Delta,\ell}(u,v) = v^{\frac{1}{2}(\Delta_2+\Delta_3)} g^{\mathcal{O}_1\mathcal{O}_2;\mathcal{O}_3\mathcal{O}_4}_{\Delta,\ell}(u,v) \pm (u \leftrightarrow v),
\tag{40}
$$

where the dependence of the conformal block $g^{\mathcal{O}_1\mathcal{O}_2;\mathcal{O}_3\mathcal{O}_4}_{\Delta,\ell}(u,v)$ on the dimensions $\Delta_a$ of the operators $\mathcal{O}_a$, $a = 1,\ldots,4$, comes only through the combinations $\Delta_1 - \Delta_2$ and $\Delta_3 - \Delta_4$.

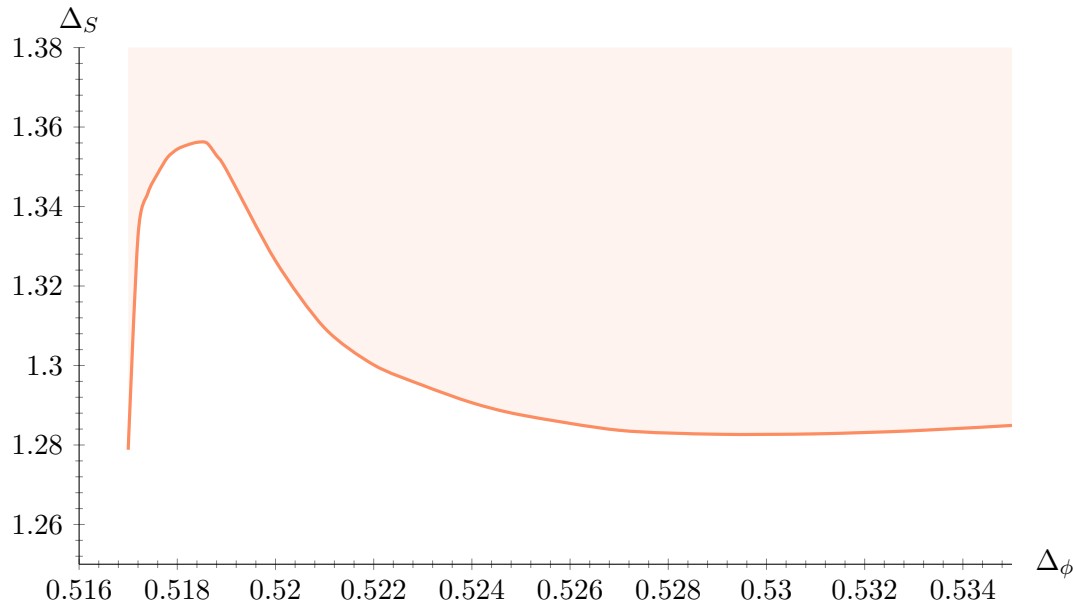

Figure 4: Upper bound on the dimension of the first singlet operator $S$. For this plot we assume that $\Delta_X$ lies on the bound of Fig. 1 and we impose the gaps $\Delta_{X'} > 3.0$ and $\Delta_{\phi'} > 1.0$. The red area is excluded.

# 3 Results

In this section we present the results of our numerical exploration of $\phi$-$X$ mixed correlators in unitary theories with cubic symmetry. Parameter choices in `PyCFTBoot` are as follows: `nmax=7`, `mmax=5`, `kmax=32`. We include spins up to $\ell_{\max} = 26$. For SDPB we use the options `-findPrimalFeasible` and `-findDualFeasible`,[4] and we further set `-precision=660`, `-dualErrorThreshold=1e-20`, and default values for other parameters.

As a first result we would like to mention that the plot of Fig. 1 remains the same even when we use the system of $\phi$-$X$ mixed correlators. Assuming that $\Delta_X$ lies on the bound of Fig. 1, we can obtain a bound on $\Delta_S$ using the additional assumptions $\Delta_{X'} > 3.0$ and $\Delta_{\phi'} > 1.0$. We remind the reader that $\phi$ appears in the $\phi \times X$ OPE in the $Y'^+$ set of operators for spin zero, and the corresponding OPE coefficient is equal to the OPE coefficient with which $X$ appears in the $\phi \times \phi$ OPE, i.e. $c_{\phi\phi X} = c_{\phi X\phi}$. Unless otherwise noted, we use this OPE coefficient equality throughout this section. The next scalar operator in the $Y'^+$ set of operators is here called $\phi'$, and we impose the mild gap $\Delta_{\phi'} > 1.0$ in order to ensure that the equality of the OPE coefficients we mentioned provides an actual constraint [18].[5] The bound is shown in Fig. 4. The bound of Fig. 4 clearly indicates a region, located horizontally around $\Delta_\phi = 0.518$, in which one could expect to find a special solution to the crossing equations.

Still assuming that $\Delta_X$ lies on the bound of Fig. 1, we now obtain a region of allowed $\Delta_S$, with the assumptions $\Delta_{S'}, \Delta_{X'} > 3.0$, and $\Delta_{\phi'} > 1.0$. Our results are shown in Fig. 5. Compared to Fig. 4, the only extra assumption for the plot of Fig. 5 is that we take $\Delta_{S'} > 3.0$. Clearly, the shape of the allowed region in Fig. 5 suggests that something special is happening in the region of the fist. This is because of the fact that the narrowing of the allowed region observed all the way to the wrist does not continue, but instead we see a widening. This

---

[4]With these options if SDPB finds a primal feasible solution then the assumed operator spectrum is allowed, while if it finds a dual feasible solution then the assumed operator spectrum is excluded.

[5]We have verified that our results are not sensitive to the gap we choose on $\Delta_{\phi'}$, assuming it remains between 0.6 and 1.1.

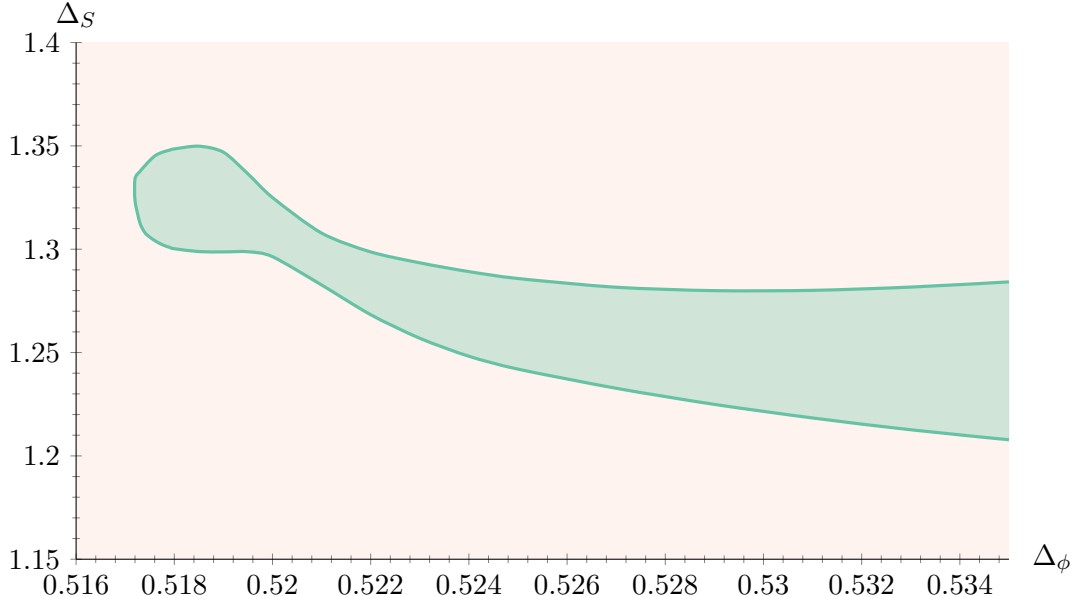

Figure 5: Allowed region, in green, for the dimension of the first singlet operator $S$. The red area is excluded. To obtain this plot we make the assumptions $\Delta_{S'}, \Delta_{X'} > 3.0$ and $\Delta_{\phi'} > 1.0$. The allowed region here looks like an arm, with the narrow wrist and the wider fist.

indicates that a solution to crossing symmetry, and hence a CFT, lies in the fist of the allowed region.

Let us now make the assumptions $\Delta_{S'} > 3.8$, $\Delta_{X'} > 3.0$, and $\Delta_{\phi'} > 1.0$. We only raised the gap of $S'$ compared to the previous choices. We can see that, as we raise this gap, the wrist of the allowed region in Fig. 5 narrows. With $\Delta_{S'} > 3.8$ the fist actually separates from the rest of the arm, and we obtain an isolated allowed region! This is shown in Fig. 6. Note that had we not imposed the equality $c_{\phi\phi X} = c_{\phi X\phi}$, we would not have obtained the separation that led to the isolated allowed region. We would like to point out that an isolated region of roughly the same size and shape is obtained for choices of the $\Delta_{S'}$ gap between $\Delta_{S'} \approx 3.7$ and $\Delta_{S'} \approx 3.9$. For $\Delta_{S'} \gtrsim 3.9$ the isolated allowed region is abruptly lost. This is consistent with results of [4, Fig. 7]. The isolated regions for the different choices of the $\Delta_{S'}$ gap are shown in Fig. 7.

Figs. 6 and 7 contain the most important results of this paper. It is natural to assert that the isolated allowed regions seen there contain operator dimensions of an actual strongly-coupled CFT.

## 4  Conclusion

In this work we have carried out a detailed numerical analysis of theories with cubic symmetry in three dimensions. We analyzed a system of mixed four-point functions, and after experimenting with assumptions on the spectrum we managed to find an isolated region allowed by unitarity and crossing symmetry. Based on earlier bootstrap experience, where similar allowed regions were found around already known CFTs, we concluded that our allowed region hosts a CFT with cubic symmetry. This CFT has only one relevant scalar singlet operator. In experiments this would correspond to the temperature that needs to be tuned in order to reach the critical point.

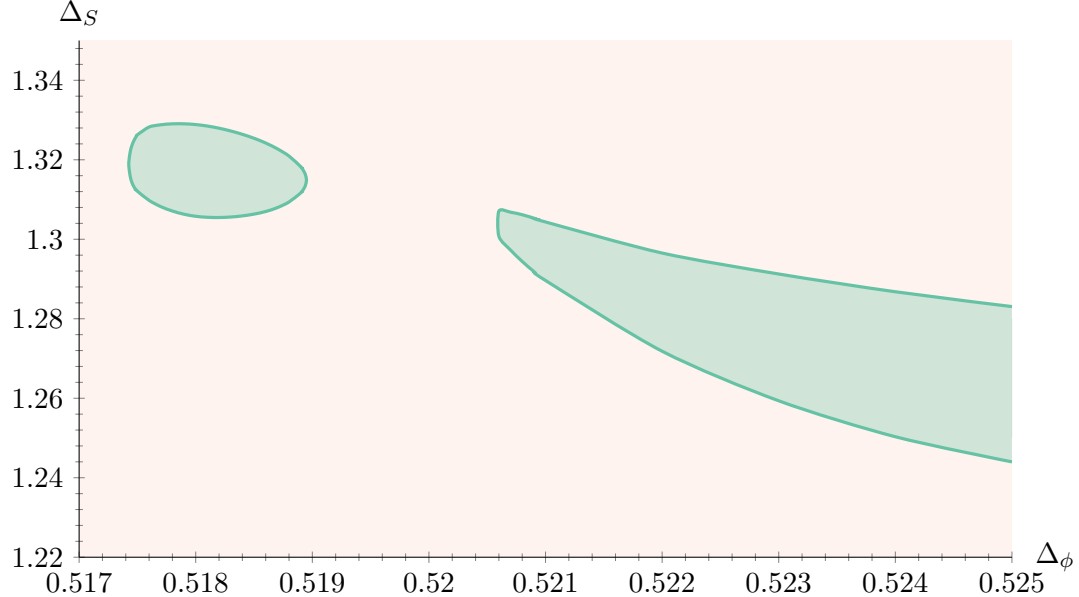

Figure 6: Allowed region, in green, for the dimension of the first singlet operator $S$. The red area is excluded. To obtain this plot we make the assumptions $\Delta_{S'} > 3.8$, $\Delta_{X'} > 3.0$ and $\Delta_{\phi'} > 1.0$.

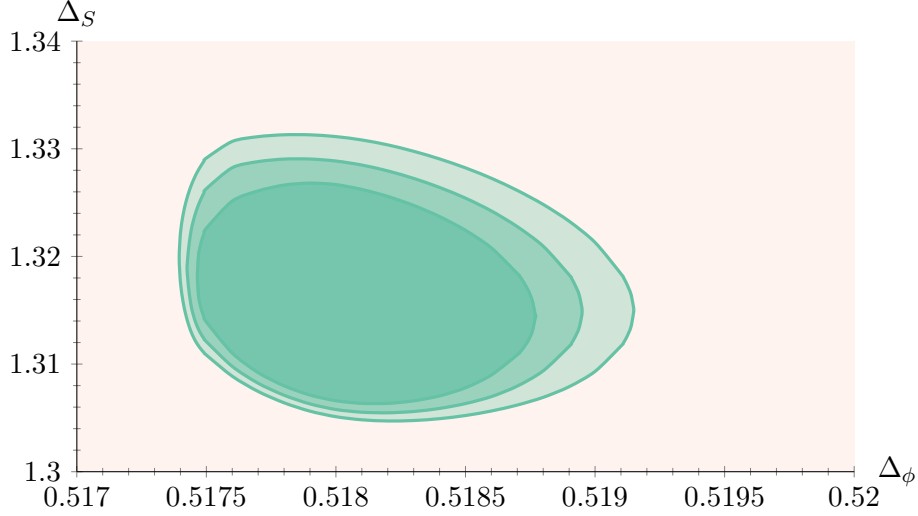

Figure 7: Allowed regions, in green, for the dimension of the first singlet operator $S$. The red area is excluded. Three closed allowed regions are plotted in green. For all of them we assume $\Delta_{X'} > 3.0$ and $\Delta_{\phi'} > 1.0$. We also assume $\Delta_{S'} > 3.7, 3.8, 3.9$ to obtain the largest, intermediate, and smallest allowed region, respectively.

The CFT we are studying here does not appear to be previously known. The standard tool for looking for CFTs in $d = 3$ is the $\varepsilon$ expansion below $d = 4$. The $\varepsilon$ expansion gives a nontrivial cubic theory in $d = 4 - \varepsilon$ [1, 24, 25], but as already noticed in [4] results for operator dimensions obtained for that theory are in disagreement with what we find here.[6] For example, the $\varepsilon$ expansion [28] as well as the fixed-dimension methods used in [29] give $\Delta_{S'}$ just slightly above marginality, which is not the case for our theory. We find the possibility that the $\varepsilon$ expansion produces wrong results unlikely, although we cannot exclude it.

---

[6]The results of [4] were obtained with a spectrum analysis [26, 27] consistent with our results in this work.

Another, more likely possibility, is that the theory we find here is a theory with cubic symmetry that cannot be captured with perturbative methods. A class of such theories can be obtained after resummations of perturbative beta functions that lead to extra fixed points not found in perturbation theory. As an example, these resummation methods have suggested a non-perturbative fixed point for $O(3) \times O(2)$ frustrated spin systems [30–32], evidence for which has also been found with the bootstrap [33]. The existence of such fixed points, however, has been challenged in [34–36]. Using the results of [29] a non-perturbative fixed point cannot be found for cubic theories.[7] It is possible that resummations performed with higher than six-loop results would reveal evidence for a strongly-coupled cubic CFT not seen in the $\varepsilon$ expansion. Clearly, Monte Carlo simulations would be very helpful in settling this issue. Note that our CFT has no relevance to cubic magnets at the critical point, for critical exponents for those systems have been measured and are very close to the ones predicted by the Heisenberg model. We propose to refer to our CFT by the name "Platonic CFT".[8]

Future work includes enlarging the set of operators we consider in our four-point functions. In this work we considered two operators, namely $\phi_i$ and $X_{ij}$, but in the future we would like to analyze numerically the system of correlators involving the scalar singlet $S$ as well. With the full system of crossing equations we hope to be able to get a three-dimensional isolated allowed region in the plot of allowed dimensions of $\phi_i$, $X_{ij}$, and $S$. We also hope to be able to push the precision in order to obtain more accurate determinations of the critical exponents. In that regard, the so-called $\theta$-scan explored in [37,38] may also lead to improvements. For these future endeavors we would greatly benefit from faster numerical optimization algorithms than are currently available.

# Acknowledgements

We would like to thank C. Behan for a modification of PyCFTBoot that allowed us to easily impose equality of OPE coefficients. AS thanks A. Vichi for many illuminating discussions. We also thank H. Osborn, S. Rychkov, and A. Vichi for comments on the manuscript. SRK would like to thank the Crete Center for Quantum Complexity and Nanotechnology for use of the Metropolis cluster, as well as G. Kapetanakis for computing support. SRK would also like to thank ITCP Crete for financial support as well as CERN-TH and AS for hospitality during a visit. The numerical computations in this paper were run on the LXPLUS cluster at CERN and the Metropolis cluster at the Crete Center for Quantum Complexity and Nanotechnology.

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
