# Peer review of "Bootstrapping Mixed Correlators in Three-Dimensional Cubic Theories"

_SciPost Physics, doi:SciPost Phys. 6, 035 (2019)_

## Round 3 · Referee Report · Anonymous (Referee 1) · 2019-2-20

Strengths

  1. The paper contains significant, possibly groundbreaking results.
  2. The paper is very clearly written.

Weaknesses

  1. If I had to give one I would say that since the paper is a straightforward application of existing methodology, to compensate for this perhaps more space could have been given to reviewing and discussing existing literature on fixed points with cubic symmetry.

Report

This is a sharp, to the point, nice paper applying conformal bootstrap methods to uncover properties of possible fixed points with cubic symmetry. The main result is the discovery of an island in the space of allowed theories, which suggests the possible existence of a new, hitherto unnoticed CFT with cubic symmetry, dubbed "Platonic CFT". Numeric results seem to be in agreement with measurements of critical exponents for certain structural phase transitions.

I have a few very minor comments given below.

Requested changes

  1. Could the authors clarify (for completeness) what is meant by the decoupled Ising model.
  2. The long sentence starting just above 1.2 and finishing after 1.3 could be improved a bit in clarity.
  3. In 2.1, it's a bit odd at first to say that $\phi_i$ lives in the off-diagonal irrep when this has just been explained as coming from a piece of the symmetric traceless tensor, in the sense that one expects that $\phi_i$ should be coming from the fundamental of O(3). I'm guessing the fundamental of O(3) turns out to transform in the same irrep thanks to eqn. 2.3? If so, might be worth making this more explicit.
  4. In 2.7, it is not obvious what irrep $Y'_l$ belongs to, given the notation up to that point. Perhaps the Z in 2.3 could be changed to a Y with a single index, or the Y in 2.7 could be changed to a Z?
  5. It would be important for the authors to comment on the dependence of their numerical results on the gap assumption on $\phi'$.
  6. It would be useful, if possible, to place the eps-expansion cubic fixed point on the exclusion plots.

  • validity: top
  • significance: high
  • originality: high
  • clarity: top
  • formatting: excellent
  • grammar: excellent

Author:  Andreas Stergiou  on 2019-03-07  [id 461]

(in reply to Report 1 on 2019-02-20)

We thank the referee for his/her suggestions. To address the referee's requested changes we have made the following additions/clarifications (please see v4 of our manuscript at https://arxiv.org/abs/1810.10015v4):

  1. We added footnote 1.
  2. We improved the wording around equations (1.2) and (1.3).
  3. Indeed (2.3) allows us to identify the fundamental and the off-diagonal irreps. We hope that our sentence before (2.3) makes that clear.
  4. Indeed the $Y'$ of (2.7) transforms in the off-diagonal irrep. We gave it the name $Y$ to make that clear. To be more explicit we added a sentence under (2.7).
  5. We added footnote 5.
  6. The $\varepsilon$ expansion cubic theory is excluded by our assumption that we sit on the bound of Fig. 1. We added in Fig. 1 the location of the cubic theory in the epsilon expansion, as given at $\varepsilon^2$ order which is the best known result to date, and we added a comment at the end of the first paragraph of page 3.

Hopefully our last amendment also helps the reader make contact with existing literature of the $\varepsilon$ expansion in theories with cubic symmetry.

---

## Editorial Decision

published